# A Randomized, Double-Blind, Placebo-Controlled Trial to Evaluate the Effects of Multi-Strain Synbiotic in Patients with Functional Diarrhea and High Fecal Calprotectin Levels: A Pilot Study

**DOI:** 10.3390/nu14235017

**Published:** 2022-11-25

**Authors:** Susie Jung, Kwang-Min Kim, Sung-Min Youn, Kyu-Nam Kim

**Affiliations:** 1Department of Family Practice and Community Health, Ajou University School of Medicine, Suwon 16499, Republic of Korea; 2GC Genome, Youngin 16924, Republic of Korea

**Keywords:** functional diarrhea, fecal calprotectin, multi-strain synbiotic, *Lactobacillus*, *Bifidobacterium*, fructooligosaccharides

## Abstract

Synbiotics, including probiotics and prebiotics, are useful for patients with functional bowel disorders. However, which synbiotics are beneficial for patients with which diseases, especially those with functional diarrhea (FDr) with high fecal calprotectin levels, is currently unknown. FDr is an extension of irritable bowel syndrome with diarrhea (IBS-D). Although fewer studies have been conducted on FDr compared to IBS-D, its importance is increasing as its prevalence increases. The aim of this study was to evaluate the effects of a synbiotic containing a mixture of *Lactobacillus* and *Bifidobacterium* and its substrate, fructooligosaccharide, on bowel symptoms, fecal calprotectin levels, fecal microbiota, and safety in FDr patients with high fecal calprotectin levels. Forty patients were randomly assigned to either a synbiotic group or a placebo group. A total of 20 subjects in the synbiotic group and 19 subjects in the placebo group completed the study (8 weeks). Changes in FDr symptoms, fecal calprotectin levels, and gut microbiota were assessed during the intervention period. At 4 and 8 weeks, the number of bowel movements tended to increase in the synbiotic group, with a significant increase in the number of formed stools rather than loose stools (*p* < 0.05). Bowel movement satisfaction was significantly increased in the synbiotic group, but not in the placebo group. Intestinal flora analysis revealed that *Lactobacillales* at the order level was increased only in the synbiotic group at the end of the intervention. In contrast, at week 8 of the intervention, log-transformed fecal calprotectin levels were significantly decreased in the synbiotic group, although the change was not significantly different from that of the placebo group. These findings suggest that the intake of a multi-strain-containing synbiotic for 8 weeks could improve gut symptoms and the intestinal microenvironment of FDr patients with high fecal calprotectin levels.

## 1. Introduction

Functional diarrhea (FDr) is defined as having more than 25% loose or watery stools (Bristol stool type 6 or 7) beginning at least 6 months ago and lasting for the last 3 months but without predominant abdominal pain or bothersome bloating [1]. Thus, patients who satisfy the criteria for irritable bowel syndrome with diarrhea (IBS-D) should be excluded from a diagnosis of FDr. Changes in the diagnostic criteria from Rome III to Rome IV (the ambiguous phrase “abdominal discomfort” in IBS was removed, the minimum pain frequency threshold was increased, and the frequency of loose stools in diarrhea was lowered from 75% to 25%) has reduced the prevalence of IBS by half (9.2% to 4.6%) [2]. According to a recent global prevalence survey, the prevalence of FDr is 4.7%, accounting for more than one-tenth of all functional gastrointestinal disorders (FGIDs). This prevalence was higher than the prevalence of IBS (4.1%) in the survey [3]. Therefore, FDr is one of the functional digestive diseases that often lead to hospital visits in primary care settings.

Although the pathophysiology of FDr is not clearly understood, emerging evidence suggests that altered gut microbiota might play an important role in the pathogenesis of FDr [4]. Alterations in gut microbiota are also associated with IBS-D. Regarding altered gut microbiota, decreased microbial diversity and richness have been found in IBS-D [5,6]. A recent meta-analysis showed that *Lactobacillus*, *Bifidobacterium*, and *Faecalibacterium prausnitzii* were reduced in patients with IBS-D compared to healthy controls or IBS patients with constipation [7].

Changes in the gut microbiota of the colon contribute to an increase in microinflammation. This was clinically confirmed by testing fecal calprotectin levels, a microinflammation indicator. Calprotectin is a protein released by leukocytes and other inflammatory cells, and it is secreted into the intestine during inflammation [8]. Patients with IBS-D and FDr, which are associated with altered gut microbiota in the colon, showed high fecal calprotectin levels [9,10]. To date, many studies have been conducted on bacterial imbalance and microinflammation of the colon in patients with IBS. However, few studies have been conducted on patients with FDr and high fecal calprotectin levels.

As FDr is considered to exist within a spectrum rather than an entity independent of IBS [1,11], we hypothesized that a synbiotic (LactominPlus^®^) formulation containing a mixture of *Lactobacillus* and *Bifidobacterium* and its substrate, fructooligosaccharide (FOS), could help improve symptoms and the intestinal microenvironment of FDr. Therefore, the purpose of this study was to investigate the effects of this synbiotic agent on bowel symptoms, fecal calprotectin, fecal microbiota, and safety in FDr patients with high fecal calprotectin levels.

## 2. Materials and Methods

### 2.1. Sample Size Estimation

As a result of assuming that the group difference (mean ± standard deviations) for the amount of change in stool frequency was −0.5 ± 0.5 times referring to a previous study [12] for calculating the number of patients, the sample size was calculated to be 32 subjects. Assuming an α-level of 0.05 (2-tailed), a power of 0.80, and a dropout rate of 20%, 20 patients were needed for each group. 

### 2.2. Study Design and Study Population

Screening was conducted for patients who complained of loose stools or watery diarrhea among patients who visited Ajou University Health Promotion Center and the Department of Family Medicine and Community Health from June 2020 to October 2021. At the first visit (week -1), medical interviews and screening tests were conducted to identify those who met the inclusion criteria (Appendix A). The inclusion criteria included satisfying the Rome IV FDr criteria. According to the Rome IV criteria, patients with symptoms for 6 months and those in which more than 25% of all stools were loose or watery, corresponding to Bristol scale types 6 and 7, for 3 months without feeling severe abdominal pain or bloating during defecation were selected [1]. 

During the screening period, demographic characteristics, vital signs, medical/surgical history, and drug intake were investigated and fecal calprotectin and fecal microbiota tests were performed. We reviewed medical records for gastroduodenoscopy or colonoscopy for organic gastrointestinal disease within the last 1 or 2 years, respectively. Hypertension, dyslipidemia, and diabetes were also confirmed through medical records or interviews. Alcohol consumption was assessed in grams of ethanol consumed per week using a step-by-step frequency method [13]. Forty-five patients with FDr were screened. Of them, 5 were excluded for the following reasons. One met the exclusion criteria, one did not consent to participate, and three had fecal calprotectin values < 11.5 mg/kg. Forty patients judged to be eligible were assigned a “random number” in the order in which they were enrolled in the study at the second visit (week 0). Patients were randomized to the placebo (20 patients) or synbiotic group (20 patients) using a computer-generated block random list with a 1:1 allocation (Figure 1).

In the case of the synbiotic group, one sachet containing a multi-strain synbiotic was ingested with sufficient water twice a day. The placebo group was given one sachet of placebo twice a day with sufficient water. During the study period, the intake of other types of probiotics, prebiotics, antibiotics, H2 blockers, and proton pump inhibitors was prohibited. The primary outcome of this study was the assessment of the effects of the synbiotic on daily intestinal symptoms compared to the placebo. Therefore, daily bowel symptoms such as bowel movement frequency, number of diarrhea events, stool hardness (confirmed by the Bristol stool scale), and bowel satisfaction were recorded (see details in 2.5.1. Assessment tools of bowel symptoms). The patients visited the hospital at week 4 (third visit) and week 8 (last visit). At the 8-week visit, fecal calprotectin, fecal microbiota, and blood tests were performed. During the 8-week study period, one person in the synbiotic group was lost to follow-up. Finally, 19 subjects in the synbiotic group and 20 subjects in the placebo group were analyzed for the study.

### 2.3. Synbiotic Preparation

Synbiotic (LactominPlus^®^) manufactured by NOVAREX Co., Ltd. (Cheongju-si, Korea) in a powder form was stored at room temperature below 25 °C. It was packaged at 6000 mg per pack and contained multiple probiotic strains (*Lactobacillus acidophilus* La-14, *Lactobacillus plantarum* Lp-115, and *Bifidobacterium animalis* subsp. *lactis* CBG-C10) (CTCBIO Inc. Gyeonggi-do, Korea) (10%, 600 mg) and fructooligosaccharide (20%, 1200 mg). The daily dosage of the synbiotic was ≥ 1 × 10^8^ CFU/day including *L. acidophilus* ≥ 2.9 × 10^7^, *L. plantarum* ≥ 4.7 × 10^7^, and *B. animalis* subsp. *lactis* ≥ 2.4 × 10^7^ CFU/day. The placebo was composed of vegi-cream (69.5%, 4170 mg/pack), yogurt-flavored cotton (0.5%, 30 mg/pack), dextrin (24.7%, 1482 mg/pack), crystalline cellulose (5.0%, 300 mg/pack), and gardenia blue color (0.3%, 18 mg/pack). It was not different from the synbiotic in taste, color, or flavor.

### 2.4. Informed Consent and Ethical Approval

All patients provided informed consent prior to enrollment in the study. This study was conducted in accordance with the Declaration of Helsinki. The study protocol was approved by Ajou University Hospital Ethics Committee (approval number: AJIRB-MED-FOD-20-064). Clinical trial numbers were obtained from Clinical Research Information Services (CRIS Registration Number: KCT0007564).

### 2.5. Measurements

#### 2.5.1. Bowel Symptom Assessment Tools

Intestinal symptoms were recorded daily during the study period in an E-diary (a program accessible by a smartphone or computer) by the patient. The intestinal symptoms evaluated included stool frequency (the number of bowel movements per day), loose stool (the number of Bristol stool type 6 or 7 shaped stools per day), formed stool (value obtained by subtracting loose stool from stool frequency), and self-reported bowel movement satisfaction (0–100 points, with higher numbers indicating higher satisfaction). All were evaluated by the patient. The intestinal symptom scores were analyzed by calculating the average value of 4 weeks at the 4th and 8th weeks of the daily survey.

#### 2.5.2. Fecal Calprotectin and Microbiology Assays

Specimens for fecal calprotectin and fecal microbiome were obtained at study enrollment (first visit) and at the end of the study (last visit, week 8). The first fecal container was distributed at the first visit (week -1) and fecal calprotectin values were examined before the start of treatment (the second visit). In this study, fecal calprotectin before treatment was expressed as week 0. For fecal calprotectin analysis, a calprotectin fluorescence enzyme immunoassay (FEIA) kit (Phadia, Uppsala, AB, Sweden) was used with an ImmunoCAP 250 (R) (Aloka, Tokyo, Japan) instrument at the Institute of Applied Technology for Green Cross LabCell (Yongin, Korea). Fecal calprotectin levels were quantified as mg/kg of feces. Values less than 11.5 mg/kg were not measured. Patients with fecal calprotectin levels of 11.5 mg/kg or more were included in this study. 

Fecal microbiome analysis was performed by GC Genome Corp. (Yongin, Korea) using a Chemagic DNA Stool Kit (PerkinElmer Inc., Waltham, MA, USA). Stool suspension (600–800 µL) was used for DNA extractions. DNA concentrations were determined fluorometrically on a Qubit^®^ 4.0 Fluorometer (Thermo Fisher Scientific, Waltham, MA, USA) using the QubitTM dsDNA HS Assay Kit. Based on these values, the DNA was diluted with nuclease-free water. The prepared DNA samples were used for 16S library construction using the NEXTflex 16S V4 Amplicon-Seq (BioO Scientific, Austin, TX, USA). The prepared library was checked with a 4200 Tape Station System (Agilent Technologies, Santa Clara, CA, USA). The library was diluted to an equimolar concentration and samples with different barcode sequences were pooled together. Paired-end sequencing was performed with the Miseq reagent kit v2 standard using a MiSeq instrument according to the manufacturer’s instructions (Illumina, San Diego, CA, USA). For measuring the overall quality of the Illumina MiSeq paired-end (PE, 2 × 250 nt) sequencing runs, 12% PhiX DNA (Illumina) was used.

### 2.6. Statistics

An independent t-test was used for comparing continuous variables between groups and a paired t-test was used to compare each group at weeks 0, 4, and 8. For categorical variables, Fisher’s exact test or the chi-squared test was performed. The distribution of fecal calprotectin values was right-skewed and analyzed as natural log values. For intestinal symptom scores and fecal calprotectin levels, the linear mixed-effect model was used to analyze the group, week, and the interaction of group and week as a random effect and fixed effect to analyze the difference between groups and before and after comparison within the groups.

The Shannon index and the weighted/unweighted UniFrac distance matrix were used to confirm the alpha diversity of the microbiome between the placebo group and the synbiotic group before and after the intervention. Linear discriminant analysis effect size (LEfSe) was performed to analyze significant differences corresponding to a *p*-value of <0.05 at the strain level. A linear discriminant analysis (LDA) score of 2 or higher was used as an index of the effect size. Advance between microorganisms was inferred based on Spearman-based reads of the strain level. The Wilcoxon test was used to determine significance. For all statistics, statistical significance was considered for *p*-values of less than 0.05. SAS version 9.4 (SAS Institute, Inc., Cary, NC, USA) was used for all statistical analyses.

## 3. Results

### 3.1. Baseline Characteristics

There were no significant clinical differences in smoking status, alcohol intake, or the prevalence of hypertension, diabetes mellitus, or dyslipidemia between the two groups (Table 1). The mean patient age was 49.8 ± 2.1 years in the synbiotic group and 46.3 ± 2.6 years in the placebo group. There was no significant difference in BMI (kg/m^2^) between the two groups (26.0 ± 1.9 vs. 26.1 ± 1.0). The mean fecal calprotectin level was 111.5 ± 27.2 mg/kg in the synbiotic group and 217.5 ± 70.0 mg/kg in the placebo group, with no significant difference between the two groups when these values were log-transformed (Table 1).

### 3.2. Intestinal Symptoms before and after Intervention

At baseline, the synbiotic group had a higher rate of loose stools as stool forms than the placebo group. At 4 and 8 weeks, the number of bowel movements tended to increase (estimate: +0.2) in the synbiotic group. There was a statistically significant increase in the number of formed stools (estimate: +0.5 and +0.4; *p* = 0.009 and *p* = 0.028, respectively), compared to loose stools. There was no significant change in stool frequency, but loose stool frequency was increased at 4 weeks after the intervention in the placebo group (Table 2 and Figure 2).

Patients’ bowel movement satisfaction increased in the synbiotic group with increases in formed stools. Satisfaction increased as the duration of synbiotic administration increased from 4 to 8 weeks (within-group *p* < 0.001). There was no difference in bowel movement satisfaction in the placebo group at 4 or 8 weeks (estimate: +15.4 and +14.7; *p =* 0.024 and *p* = 0.029, respectively) (Table 2).

### 3.3. Between and Within-Group Fecal Calprotectin Levels 

Table 3 shows a comparison of log-transformed fecal calprotectin values in the two groups at baseline and after 8 weeks of intervention. At week 8, the levels of log-transformed fecal calprotectin were decreased in both groups (*p =* 0.006 in the synbiotic group and *p* = 0.008 in the placebo group) compared to baseline. However, there was no statistically significant difference in the log-transformed fecal calprotectin changes before and after the intervention between the two groups (estimate: −0.06; *p* = 0.889).

### 3.4. Fecal Microbiota Analysis 

In the LEfSe analysis performed to identify microorganisms with significant changes in the intestinal microbiota, the multi-strain synbiotic group showed a statistically significant increase in *Lactobacillales* at the order level (*p* = 0.044) at 8 weeks, whereas no microorganisms showed a significant change at any taxonomic level in the placebo group (Figure 3a). In the alpha-diversity analysis evaluating the diversity of the intestinal microbiota, there was no significant change in either group (synbiotic and placebo group; *p* > 0.05 and *p* > 0.05, respectively) (Figure 3b).

### 3.5. Safety of the Synbiotic and Placebo

In both the synbiotic group and the placebo group, white blood cell counts, renal function tests, and liver enzyme levels were maintained within their normal ranges at week 8 of intervention, showing no statistically significant difference before and after the study (Table 4). Although there was a statistically significant difference in hemoglobin levels before and after the study in the placebo group, it was within the clinically normal range.

## 4. Discussion

This double-blind randomized study investigated the effects of a multi-strain synbiotic compared to placebo on bowel symptoms, changes in the intestinal microenvironment, and safety in adults with bowel discomfort and watery or loose stools after 8 weeks of intervention. The treatment of FDr patients with high fecal calprotectin levels with the multi-strain synbiotic improved the degree of formed stool and increased patient self-reported bowel movement satisfaction with decreased fecal calprotectin levels. In gut microbiota analysis, the synbiotic group showed an increase in *Lactobacillales* at the order level after the intervention compared to before. However, there were no significant changes in gut microbiota in the placebo group. The three strains contained in the multi-strain probiotic used in this study were *L. acidophilus*, *L. plantarum*, and *B. animalis* subsp. *latics*. Both *L. acidophilus* and *L. plantarum* belong to the order *Lactobacillales*. These findings suggest that ingested strains could settle in the gut microbiome and change its composition.

FDr and IBS are both functional bowel diseases. Although their pathophysiological mechanisms are still unclear, abnormal gut motility, visceral hypersensitivity [14], brain–gut axis alteration [15], low-grade inflammation [16], and changes in intestinal microflora [17] have been suggested. Relatively more studies on alterations in the gut microbiota have been conducted on IBS compared to FDr. Some studies showed that probiotics could improve diarrhea symptoms. In particular, some strains of *L. plantarum* and *Bifidobacterium animalis spp*. known to be effective in improving diarrhea and IBS-D. For example, *L. plantarum* 299v is known to be effective in controlling abdominal bloating [18], and *Bifidobacterium animalis* spp. XLTG11 is known to be effective in controlling antibiotic-related diarrhea [19]. Unfortunately, there has been no direct comparative study of whether combinations of probiotics are more effective than a single strain, and conclusions vary from disease to disease and from strain to strain. Until now, it has been difficult to conclude whether a single strain was effective or mixed strains were effective, and the results have been controversial. However, a recent systemic review of probiotics for treating IBS reported that a combination of probiotics was more effective in improving IBS symptoms than single strains [20]. This systemic review of 4 RCT trials revealed that when a combination of probiotics containing *Lactobacillus* and *Bifidobacterium* was administered at a dose of 10^10^ (CFU/day), it was effective in alleviating IBS symptoms, resulting in symptom relief of about 50% [20]. The most effective probiotic for IBS-D was a combination formulation containing four *Lactobacillus* species, three *Bifidobacterium* species, and one *Streptococcus* species [21]. As a prebiotic, FOS was shown to alter the gut microbiome composition to exhibit bifidogenic properties [22], and most strains of *Bifidobacterium* and *Lactobacillus* can utilize FOS [23]. FOS was reported to alleviate the symptom severity of those with functional bowel disorders [24]. It is known to have immunomodulation and anti-inflammatory effects [25,26]. Our results were consistent with these findings.

Whether multi-strain probiotics will provide host benefits through different mechanisms of action [27] or interfere with each other’s effects by intra-strain antagonism [28] remains unclear. In our previous study, when a single-strain probiotic containing *L. plantarum* was given to patients with FDr, symptoms were improved by reducing fecal calprotectin, thus improving intestinal microinflammation [29]. In the present study, when a multi-strain synbiotic containing *L. plantarum* was used together with a prebiotic for 8 weeks, fecal calprotectin levels were significantly decreased compared to those at baseline. These findings suggest that *L. plantarum* might be efficient for regulating inflammation via the regulation of IL-6, an inflammatory cytokine when it is given with mixed strains [30].

Although the exact mechanism by which multi-strain synbiotics are helpful in bowel movement satisfaction is currently unknown, it might be due to γ-aminobutyric acid (GABA) production by *Lactobacillus* and *Bifidobacterium* spp. [31,32,33]. GABA is the major inhibitory neurotransmitter in the body. As is well known, the decreased activity of central GABA is associated with depression and anxiety, and GABA is located in the enteric nerve and endocrine-like cells of the gastrointestinal tract, regulating GI tract function. The vagus nerve primarily regulates the gut–brain axis, and GABA plays a key role in activating the vagus nerve [34]. Therefore, GABA production or the activation of GABA receptors is known to be involved in physiological intestinal motility and secretion, intestinal inflammation, and immune regulation through crosstalk with the gut–brain axis [35]. The ingestion of *Lactobacillus* can modulate central GABA receptor expression via the vagus nerve, resulting in changes in emotional behavior (reductions in anxiety and depression) [36]. *Lactobacillus* and *Bifidobacterium* as psychobiotics [37] can increase GABA production. Another preclinical study on *Lactobacillus* strains suggested that GABA-producing bacteria might be involved in regulating intestinal visceral hypersensitivity [38] and vagus nerve activation. These studies might explain the mechanism by which multi-strain synbiotics, including *Lactobacillus* and *Bifidobacterium* used in our study, can increase bowel movement satisfaction after ingestion. Taken together, treatment with multi-strain synbiotics such as *Lactobacillus* and *Bifidobacterium* might not only reduce microinflammation in the colon but also increase intestinal satisfaction due to changes in the intestinal microbiota composition.

A statistically significant change in the levels of calprotectin, an indicator of colonic inflammation, was also observed before and after the intervention in the placebo group. This might be explained by the brain–gut axis theory. In other words, since functional bowel disease itself is a disease affected by psychological factors, patients’ voluntary participation in the study and self-assessment of intestinal symptoms on a daily basis might have acted as a positive factor to help improve intestinal immunity [39]. However, our finding of no significant change in gut flora in the placebo group suggests that improvements in intestinal inflammation alone might not be effective in ameliorating the symptoms of patients with FDr with high fecal calprotectin levels.

This study had several limitations. First, it was conducted at a single center. Second, the finding that the synbiotic group did not show statistically significant changes in the microinflammation of the colon or the diversity of gut bacteria in FDr patients compared to the placebo group may be due to the short study period and insufficient daily intake of FOS. Third, hematologic tests and stool analysis were performed before and after the treatment only, and whether improvements in intestinal symptoms persisted or whether there was a change in fecal microbiota during the period after the intervention were not investigated. Finally, the LEfSe analysis to find meaningful changes in gut bacteria in this study has the limitation that it cannot take into account the multivariate nature of the microbiome. However, this was the first randomized controlled trial to examine the effects of a multi-strain synbiotic in an adequate number of patients with FDr, whose prevalence has increased after the revision of the Rome criteria. In addition, it has the strength of studying various factors such as fecal microbiota and microinflammation of the colon. Furthermore, since E-diary, a method in which patients report their bowel symptoms on a daily basis, was selected, recall bias was minimized, and the reliability of the patient’s subjective symptom evaluation was high.

In conclusion, the results of this double-blind randomized controlled human study demonstrated that the daily intake of multiple strains (*L. acidophilus*, *L. plantarum*, and *B. animalis* subsp. *lactics*) and FOS for 8 weeks were effective in improving the degree of formed stool and patient-self reported bowel movement satisfaction compared to placebo in FDr patients with high fecal calprotectin levels according to the Rome IV criteria. In addition, we found a significant increase in *Lactobacillales* at the order level in the fecal microbiome analysis of the synbiotic group. Given the expected increase in the prevalence of FDr according to the Rome IV criteria for functional bowel disorders, our findings provide a basis for improving bowel symptoms in FDr patients. In the future, a large-scale study is needed to evaluate the effects and stability of *Lactobacillus* and *Bifidobacterium* mixed strains or synbiotics in patients with FDr and IBS-D, which may share a common pathophysiology.

## Figures and Tables

**Figure 1 nutrients-14-05017-f001:**
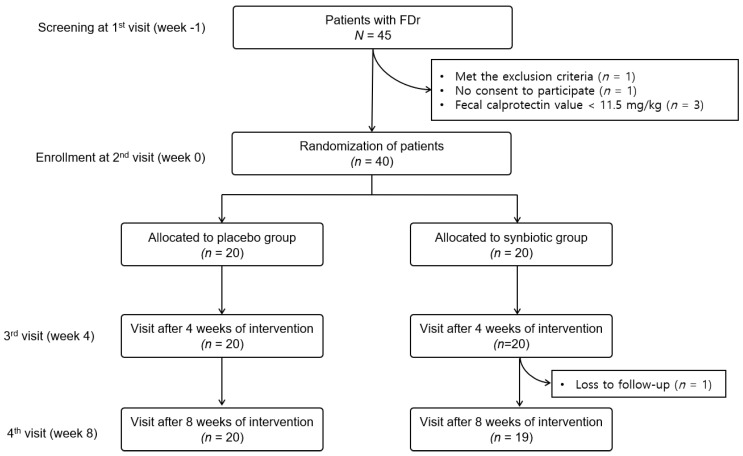
Flow diagram showing the selection of subjects for this study. FDr, functional diarrhea.

**Figure 2 nutrients-14-05017-f002:**
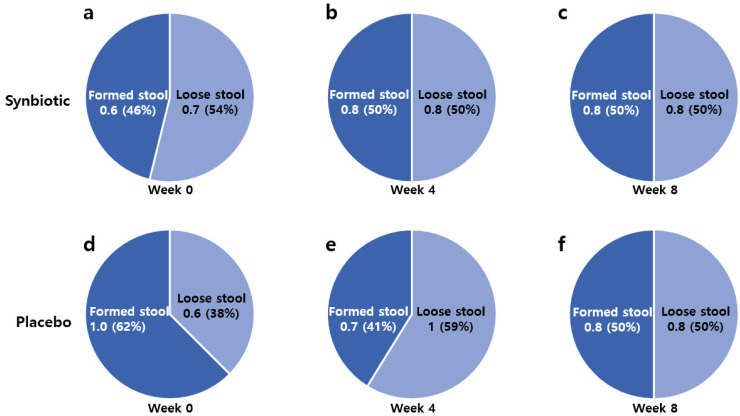
Changes in the percentage of loose and formed stools at baseline, 4 weeks, and 8 weeks after intervention. Changes in the proportion of formed and loose stools at (**a**) week 0 (baseline), (**b**) week 4, and (**c**) week 8 in the synbiotic group compared to the placebo group at (**d**) week 0 (baseline), (**e**) week 4, and (**f**) week 8.

**Figure 3 nutrients-14-05017-f003:**
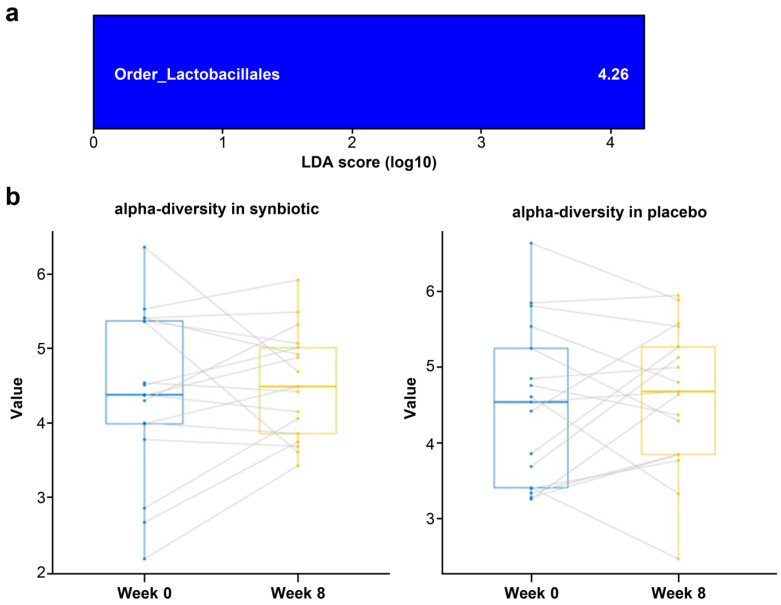
Changes in gut microbiota based on (**a**) LEfSe analysis and (**b**) the Shannon index. (**a**) Linear discriminant analysis (LDA) effect size (LEfSe) analysis in the synbiotic group. Significant results with LDA values of 4.0 or higher and *p* < 0.05 in Wilcoxon’s rank sum test are presented. (**b**) Alpha-diversity analysis measured by the Shannon index. Wilcoxon’s rank sum test was used to compare differences within groups (synbiotic and placebo group; *p* > 0.05 and *p* > 0.05, respectively).

**Table 1 nutrients-14-05017-t001:** Baseline characteristics of the study population.

	Synbiotic (*n* = 19)	Placebo (*n* = 20)	*p*-Value
Age (years)	49.8 ± 2.1	46.3 ± 2.6	0.304
Male, *n* (%)	13 (68)	14 (70)	0.915
BMI (kg/m^2^)°	26.0 ± 1.9	26.1 ± 1.0	0.178
Current smoker, *n* (%)	2 (10.5)	7 (35.0)	0.127
Ex-smoker, *n* (%)	5 (26.3)	4 (20.0)	0.717
Weekly alcohol intake	5.5 ± 2.6	5.4 ± 1.7	0.967
Fecal calprotectin (mg/kg)	111.5 ± 118.5	217.5 ± 312.6	
Log-transformed FC	4.34 ± 0.19	4.71 ± 0.25	0.254
Hypertension	4	5	1.000
Diabetes mellitus	1	2	1.000
Dyslipidemia	7	9	0.605

BMI, body mass index; FC, fecal calprotectin. All values of age, BMI, weekly alcohol intake, fecal calprotectin and log-transformed FC represent mean ± standard deviation.

**Table 2 nutrients-14-05017-t002:** Improvement in functional diarrhea symptoms after intervention.

	Synbiotic	Placebo	Estimate ^a^	*p*-Value ^b^
Stool frequency (#/day)				
Week 0	1.3 ± 0.1	1.6 ± 0.2		
Week 4	1.6 ± 0.1	1.7 ± 0.1	+0.2	0.148
Week 8	1.6 ± 0.1	1.6 ± 0.9	+0.2	0.05 *
Within-group comparison	0.002 **	0.625		
Loose stool				
Week 0	0.7 ± 0.1	0.6 ± 0.1		
Week 4	0.8 ± 0.1	1.0 ± 0.1	−0.3	0.084 *
Week 8	0.8 ± 0.2	0.8 ± 0.1	−0.1	0.387
Within-group comparison	0.658	0.093 *		
Formed stool				
Week 0	0.6 ± 0.1	1.0 ± 0.2		
Week 4	0.8 ± 0.1	0.7 ± 0.1	+0.5	0.009 **
Week 8	0.8 ± 0.2	0.8 ± 0.1	+0.4	0.028 **
Within-group comparison	0.067 *	0.205		
Bowel movement satisfaction				
Week 0	44.2 ± 4.9	45 ± 4.4		
Week 4	60.8 ± 5.0	46.5 ± 5.6	+15.4	0.024 **
Week 8	63.4 ± 5.3	49.4 ± 5.5	+14.7	0.029 **
Within-group comparison	<0.001 **	0.363		

All values of stool frequency, loose stool, formed stool, and self-reported bowel movement represent mean ± standard deviation. ^a^ Estimate is the change in the synbiotic group compared to the change in the placebo group. The linear mixed-effect model was used to analyze the effects of group x week at week 4 and week 8. ^b^
*p*-value for group × time effect. * *p* < 0.1; ** *p* < 0.05.

**Table 3 nutrients-14-05017-t003:** Changes in log-transformed fecal calprotectin at week 0 (baseline) and week 8.

	Synbiotic	Placebo	Estimate ^a^	*p*-Value ^b^
Week 0	4.34 ± 0.19 (111.5 ± 27.2) ^§^	4.71 ± 0.25 (217.5 ± 70.0) ^§^		
Week 8	3.51 ± 0.23 (56.7 ± 14.4) ^§^	3.91 ± 0.2 (73.6 ± 16.6) ^§^	−0.06	0.889
Within-group comparison	0.006	0.008		

All values of log-transformed fecal calprotectin represent mean ± standard deviation. ^a^ Estimate is the change in the test group compared to the change in the control group. The linear mixed-effect model was used to analyze the effects of group x week. ^b^
*p*-value for group × time effect. ^§^ Values in parentheses indicate fecal calprotectin values without taking the natural log transformation.

**Table 4 nutrients-14-05017-t004:** Blood parameters at baseline and 8 weeks of intervention.

	Synbiotic (*n* = 19)			Placebo (*n* = 20)		
	Baseline	8 Weeks	*p*-Value	Baseline	8 Weeks	*p*-Value
WBC (×10^3^/μL)	6.7 ± 0.6	6.4 ± 0.4	0.250	6.3 ± 0.4	6.3 ± 0.4	0.904
Hb (g/dL)	14.6 ± 0.3	14.6 ± 0.2	0.795	14.9 ± 0.3	14.3 ± 0.3	0.005
BUN (mg/dL)	13.5 ± 0.9	13.7 ± 0.9	0.841	8.8 ± 0.8	13.1 ± 0.9	0.754
Creatinine (mg/dL)	0.83 ± 0.03	0.84 ± 0.04	0.679	0.87 ± 0.03	0.84 ± 0.03	0.154
ALT (IU/L)	45.9 ± 9.1	45.4 ± 9.4	0.904	26.2 ± 3.2	29.6 ± 4.4	0.379
AST (IU/L)	35.1 ± 6.4	32.9 ± 5.5	0.506	22.7 ± 1.8	25.3 ± 2.5	0.431

All values represent mean ± standard deviation. WBC, white blood cells; Hb, hemoglobin; BUN, blood urea nitrogen; ALT, alanine aminotransferase; AST, aspartate aminotransferase.

## Data Availability

The data in this study are not publicly available but can be requested from the corresponding author.

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
