# Peer review of "A Randomized, Double-Blind, Placebo-Controlled Trial to Evaluate the Effects of Multi-Strain Synbiotic in Patients with Functional Diarrhea and High Fecal Calprotectin Levels: A Pilot Study"

_nutrients, 2022, doi:10.3390/nu14235017_

Round 1
Reviewer 1 Report (Previous Reviewer 2)
1 The answer regarding the composition of the product is unsatisfactory. The manufacturer needs to know if he measures the CFU of the individual strains or if he bases the composition on the percentage input of the individual components. This information is very important for the preparation of systematic reviews and meta-analyses that report the doses of individual strains.
2. I also noted that the data in Table 2 are not very clear. How is the ":Estimate" value calculated? It should also be stated that the values in the table are the mean ±SD. In section 2.5.1, the authors state that: The intestinal symptoms evaluated included 1/the number of bowel movements per day, 2/the number of diarrhoea or loose stools, 3/the shape of the stool evaluated by the Bristol stool scale, and 4/self-reported bowel movement satisfaction (0 - 10 points, with higher numbers indicating higher satisfaction). Meanwhile, Table 2 shows: 1/stool frequency, 2/loose stool, 3/formed stool, 4/bowel movement satisfaction. It can be seen that points 3 do not overlap and points 2 only partially overlap (no diarrhoea) which needs to be explained in the text.
Author Response
Authors’ response
Thank you for your detailed paper review. The opinions of the authors regarding your points are described in red.
1 The answer regarding the composition of the product is unsatisfactory. The manufacturer needs to know if he measures the CFU of the individual strains or if he bases the composition on the percentage input of the individual components. This information is very important for the preparation of systematic reviews and meta-analyses that report the doses of individual strains.
: Thanks for your point. We additionally conveyed this information to the manufacturer, and the manufacturer confirmed the composition of the strain as follows; The each strains (Lactobacillus acidophilus La-14, Lactobacillus plantarum Lp-115, and Bifidobacterium animalis subsp. lactis CBG-C10) were cultured and counted CFU separately (L. acidophilus La-14 ≥2.0×1011 CFU/g, L. plantarum Lp-115 ≥4.0×1011 CFU/g, and B. animalis subsp. lactis CBG-C10 ≥5.0×1011 CFU/g). The certain ratio blending (L. acidophilus La-14 1.25%, L. plantarum Lp-115 1%, and B. animalis subsp. lactis CBG-C10 0.4%) was proceeded to produce the final product (Lactomin Plus). After that, the total CFU of Lactomin Plus were measured to standardization for setting up the product standard specification.
We hope you will be satisfied with our answer.
- I also noted that the data in Table 2 are not very clear. How is the ":Estimate" value calculated?
: Estimate means the change in the placebo group for 8 weeks versus that in the probiotic group for 8 weeks. In the linear mixed-effect model, the estimate is a statistical method that compares the change in synbiotic group compared to the change in the placebo group.
Please refer to the following example.
If the values of placebo group are 1.6 at week 0 and 1.6 at week 8, the change in the placebo group for 8 weeks is 0.0 (B).
If the values of probiotic group are 1.4 at week 0 and 1.6 at week 8, the change in the LP group for 8 weeks is 0.2 (A).
Therefore, estimate is calculated as 0.2 from "0.2 (A) - 0.0 (B)".
Table 2 shows that the frequency (#/day) of formed stools in the synbiotic group increased after 8 weeks (0.6 at 0 week → 0.8 at 8 week: delta 0.2), whereas in the placebo group it decreased (1.0 at 0 week → 0.8 at 8 week: delta -0.2). Therefore, the estimate of the synbiotic group and the placebo group was +0.4 [estimate = 0.2 – (-0.2); p = 0.028], which showed that the change in formed stool in the synbiotic was statistically significant compared to the placebo. The statistical significance of the estimates for the changed values of the two groups was run through the SAS version 9.4 statistical program. We hope our explanation will help you understand.
It should also be stated that the values in the table are the mean ±SD.
: In each table legend, we added the description that the values mean ‘mean ± standard deviation’.
- In table 2. All values of stool frequency, loose stool, formed stool, and self-reported bowel movement represent mean ± standard deviation.
- In table 3. All values of log-transformed fecal calprotectin represent mean ± standard deviation.
- In table 4. All values represent mean ± standard deviation.
In section 2.5.1, the authors state that: The intestinal symptoms evaluated included 1/the number of bowel movements per day, 2/the number of diarrhoea or loose stools, 3/the shape of the stool evaluated by the Bristol stool scale, and 4/self-reported bowel movement satisfaction (0 - 10 points, with higher numbers indicating higher satisfaction). Meanwhile, Table 2 shows: 1/stool frequency, 2/loose stool, 3/formed stool, 4/bowel movement satisfaction. It can be seen that points 3 do not overlap and points 2 only partially overlap (no diarrhoea) which needs to be explained in the text.
: We agree with your points. The terms in the text are clearly matched as shown in the table, and the meanings are explained in parentheses so that readers do not misunderstand.
- included stool frequency (the number of bowel movements per day), loose stool (the number of Bristol stool type 6 or 7 shaped stools per day), formed stool (value obtained by subtracting loose stool from stool frequency), and self-reported bowel movement satisfaction (0 – 100 points, with higher numbers indicating higher satisfaction).

Reviewer 2 Report (New Reviewer)
The study is interesting and relevant, considering the absence of studies with this specific symbiotic, as far as I could verify. The manuscript is clear and the methodology is sound. However, some issues must be clarified before it can be accepted for publication:
- The symbiotic product used is not well known. Are there previous evaluations of its effectiveness? Or the strains used in its formulation? Was there any rationale for the specific choice of LactominPlus?
- Could the authors inform whether the species included in the formulation (Lactobacillus acidophilus La-14, Lactobacillus plantarum Lp-115, and Bifidobacterium animalis subsp. lactis CBG-C10) use FOS?
- Still in relation to FOS, there are studies that employ doses higher than 2.4 g/day. A comment on the discussion regarding dosage and the absence of effects on the microbiota compared to placebo would be important.
- Line 275: animals should be in italics, but spp. and the strain designation (XLTG11) do not
Author Response
Authors’ response
The study is interesting and relevant, considering the absence of studies with this specific symbiotic, as far as I could verify. The manuscript is clear and the methodology is sound. However, some issues must be clarified before it can be accepted for publication:
: Thank you for your thoughtful comments on our paper. The opinions of the authors regarding your points are described in red. The content of the manuscript changed by responses to reviewer are highlighted in blue.
- The symbiotic product used is not well known. Are there previous evaluations of its effectiveness? Or the strains used in its formulation? Was there any rationale for the specific choice of LactominPlus?
: Thank you for your comments. Although there were studies on the strains we used, there was no clinical study on the effect of a complex of these strains on patients with functional diarrhea, so this study was designed. The following table shows the study results of the strains used in this study.
|
No. |
Strain |
Division |
Contents |
Ref. |
|
1 |
L. acidophilus La-14 |
Functionality |
In a diarrhea-induced mouse model, Intake of La-14 or spent culture supernatant reduced the rate of loose stool and the number of accumulated diarrhea |
1 |
|
2 |
Pathogen suppresion |
Produced bacteriocins to inhibit the pathogen. |
2 |
|
|
3 |
L. plantarum Lp-115 |
Functionality |
In a Trinitro-Benzene Sulfonic acid(TNBS)-induced colitis Rat, Intake of sulfasalazine and LP-115 for 21 days significantly reduced MPO activity |
3 |
|
4 |
Intestinal proliferation |
Intake of Fermented milk containing 2x1011 cell significantly increased L. plantarum level compared to before taking |
4 |
|
|
5 |
B. lactis CBG-C10 |
Patent |
Isolated from the large intestine of mammals that produces CLA (conjugated linoleic acid) and the CBG-C10 is a patented in Korea(1014463090000) |
- |
- Could the authors inform whether the species included in the formulation (Lactobacillus acidophilus La-14, Lactobacillus plantarum Lp-115, and Bifidobacterium animalis subsp. lactis CBG-C10) use FOS?
: Thanks for the in-depth question. In order to practically answer your question, research should be conducted by dividing into two groups: synbiotic (probiotic + prebiotic) group and probiotic group. Unfortunately, there are no such studies using the strains used in our study. Furthermore, to date, there are few studies comparing the effects of synbiotics and probiotic bacteria alone. As far as we know, there was a study by Basturk, A., etc. (5) that could answer some of the questions about this. They were divided into synbiotic, probiotic, and prebiotic groups in children with irritable bowel syndrome, and the study was conducted for 4 weeks. Their findings indicated that the synbiotic group had superior effects on irritable bowel syndrome. In addition, generally reported that Lactobacilli and Bifidobacteria use FOS to produce fecal lactate and butrate, and decrease fecal ammonia, isobutyrate, isovalerate and total branched-chain fatty acid concentration (6). Therefore, we think that our research results can also be theoretically inferred that probiotics may have shown synergistic effects using FOS.
- Still in relation to FOS, there are studies that employ doses higher than 2.4 g/day. A comment on the discussion regarding dosage and the absence of effects on the microbiota compared to placebo would be important.
: Thank you for your comments, and we have added this content to the limitation of the discussion. “. Second, the finding that the synbiotic group did not show statistically significant changes in the microinflammation of the colon or the diversity of gut bacteria in FDr patients compared to the placebo group may be due to the short study period and insufficient daily intake of FOS”.
- Line 275: animals should be in italics, but spp. and the strain designation (XLTG11) do not
: We acknowledge our mistake and correct it according to your recommendation.
- and Bifidobacterium animalis spp. XLTG11 is known to
References
- -Y. Luo et al. Effects of spent culture supernatant of lactobacillus acidophilus on experimental diarrhea. December 2007Journal of Chinese Pharmaceutical Sciences 42(24):1908-1910.
- Todorov SD, et al. Bacteriocin production and resistance to drugs are advantageous features for Lactobacillus acidophilus La-14, a potential probiotic strain. New Microbiol. 2011;34(4):357-370.
- Paroschi TP, et al. Effects of Sulfasalazine, Lactobacillus Plantarum (Lp-115) and Fish Oil in Experimental Colitis. SM J Food Nutri Disord. 2015; 1(1): 1005.
- Costa GN, Marcelino-Guimarães FC, Vilas-Bôas GT, Matsuo T, Miglioranza LH. Potential fate of ingested Lactobacillus plantarum and its occurrence in human feces. Appl Environ Microbiol. 2014;80(3):1013-1019.
- Baştürk A, Artan R, Yılmaz A. Efficacy of synbiotic, probiotic, and prebiotic treatments for irritable bowel syndrome in children: A randomized controlled trial. Turk J Gastroenterol. 2016;27(5):439-443.
- Swanson KS, et al. Fructooligosaccharides and Lactobacillus acidophilus modify gut microbial populations, total tract nutrient digestibilities and fecal protein catabolite concentrations in healthy adult dogs. J Nutr. 2002;132(12):3721-3731.

This manuscript is a resubmission of an earlier submission. The following is a list of the peer review reports and author responses from that submission.
Round 1
Reviewer 1 Report
Main comments:
1. The main concern is the quality of the paper. English should be improved, as well as the logical continuity of the text.
2. Why is analysed preparation referred to as the ‘synbiotics’ in the plural, when there was only LactominPlus tested (e.g. line 16, 66 or title)? I would recommend using the singular form of the noun since the plural one is misleading and suggest the use of more than one synbiotic during the described study.
3. Introduction: This section covers the most relevant information; nevertheless, it could be improved by adding a description of all points of Rome IV criteria that are also mentioned in the methodology and was the baseline to select candidates for the trial.
4. Materials and methods: In general the section is written well in the matter of describing vital information. Although, in subsection 2.1 Authors already stated the number of patients in each study group, whereas the sample size estimation was not discussed until subsection 2.5. I would recommend changing the order of subsections.
5. Results: I would recommend dividing this section into subsections to make it more clear and easy to read.
6. The discussion of obtained results is well written and the summary of study limitations is an advantage. Although I see more limitations of this trial, such as the fact, that all analyses were performed only before and after the treatment, whereas during the conduction of the study the faecal or blood samples were not collected. Moreover, unfortunately, the Authors did not include a follow-up period after the intervention was finished. This could give information on how long the effect of synbiotic use could last, and if there were differences among research groups during such time.
Detailed comments:
Lines 37-38: The sentence indicates that the only symptom of IBS-D, besides diarrhoea, is abdominal pain. I think this brings an insufficient amount of information, especially, since the following sentence in which Rome criteria are mentioned also brings up only pain and diarrhoea. What about other symptoms (bloating, gases, etc.)? Can they be observed in patients diagnosed with FDr?
Line 39: ‘the ambiguous word 'abdominal discomfort' in IBS was removed’; ‘abdominal discomfort’ is a phrase, not a word.
Lines 42-45: I would recommend giving the percentage of IBS prevalence after ‘has reduced the prevalence of IBS by half’, then the sentence beginning with ‘In contrast…’ would have more sense. If such changes would be applied the sentence ‘This prevalence is higher than the prevalence of IBS (4.1%)’ could be deleted.
Lines 48, 49, 58: I would recommend finding a synonym for the word ‘alteration’
Lines 48-49: ‘FDr is currently 48 considered a disease in the continuum of IBS-D [1,5].’ The sentence is redundant here, especially, since it is repeated in lines 61-62.
Line 49: Authors should consider changing the word ‘continuum’ to ‘spectrum’.
Line 55: It supposed to be ‘micro-inflammation’ as in lines 50 and 51 or previously used form was incorrect?
Lines 58-61: Authors should consider moving sentences ‘Regarding alteration of gut microbiota, decreased microbial diversity and richness have been found in IBS-D [9,10]. A recent meta-analysis has shown that Lactobacillus, Bifidobacterium, and Faecalibacterium prausnitzii are reduced in those with IBS-D than in healthy controls or IBS patients with constipation [11].’ After ‘Alteration of gut microbiota is also associated with IBS-D. F’
Lines 70-77: Patients are mostly referred to as ‘those’ and ‘them’. Authors should consider using words such as ‘patients’ or ‘individuals’.
Line 84: The repetition of the word ‘medical’ is redundant
Line 90: What did the Authors mean by ‘sufficient water’?
Line 90: Is the word ‘pouch’ correct and scientific-like? I would recommend changing it into ‘sachet’
Line 98: Once the Authors use ‘synbiotic group’, other times ‘synbiotics group’. The Authors should use the first form.
Lines 104-105: Is it possible for Authors to give full strain specifics, such as the collection from which they were obtained and codes?
Lines 108-109: Is it possible for the Authors to indicate the exact composition of the placebo?
Lines 123-124: The sentence should be rephrased. Were the bowel symptoms assessed only during the 4 weeks of the trial?
Lines 161-164: Authors used two different abbreviations for linear discriminant analysis – LEfSe and LDA. The second one, indeed, stands for the linear discriminant analysis, whereas LEfSe is used for Linear discriminant analysis Effect Size, which was not described in the text. It is misleading for the potential reader.
Line 166: The producer of SAS should be mentioned in the text.
Line 189: The legend in figure 2 is redundant since the names of the series of data are presented on each pie chart.
Line 205: In Tables 1 and 2 Authors used only ‘Synbiotics’ in the header of the column to describe the group, whereas in table 3 – ‘the multi-strain synbiotics’. Is it necessary?
Line 209, 211: Authors should consider using the word ‘microorganisms’ instead of ‘organisms’
Line 210: Authors should consider using the word ‘microbiota’ instead of ‘microflora’.
Lines 221-229: Did the Authors analyse the differences between research groups or only before and after the treatment within each group? There are visible discrepancies in ALT and AST levels between groups, which are not mentioned.
Lines 236-238: The word ‘interventions’ was used 3 times in one sentence.
Lines 265-273: Did the Authors use the same strain of L. plantarum in both studies, the previous one with a single strain probiotic and the currently described trials? If not, these conclusions are too far-reaching.
Lines 274-286: Authors could improve this part by adding the explanation of GABA, what it stands for and what it is.
Line 311: Authors described the increase of Lactboacillales order in the faecal microbiota; therefore, it should be stated in the conclusions, that these bacteria order abundance was increased, not Lactobacillus specifically.
Reviewer 2 Report
Dear Authors,
The study is interesting but needs some important additions.
1. What exactly was the recruitment of patients, was it only for those with complaints or for all visiting medical centres?
2. On what basis was the cutoff point for faecal calprotectin concentration determined?
3. The inclusion and exclusion criteria should be highlighted, preferably in table (box) form.
4. Please provide the taxonomic names of the strains, how was the dosage of the strains determined?
5. The method of microbiota analysis does not take into account the compositional nature of microbiome data. LEfSe ignores the multivariate nature of the microbiome. This analysis requires the use of appropriate statistical tools.
6. Despite the sample size calculation, the study is a pilot study, which should be stated in the title.
7. The two elements specified in the title: 1/intestinal environment, and 2/elevated faecal calprotectin level need proper analysis - microbiota or defined - calprotectin level. Currently, the title is inadequate for the results obtained.
Reviewer 3 Report
Susie Jung et al. performed a randomized double-blind study which aimed to compare the effect of a synbiotic to that of a placebo on functional diarrhea, fecal calprotectin levels and some characteristics of the fecal microbiota at the genus level in 40 patients suffering from functional diarrhea and having detectable levels of fecal calprotectin.
Comments and questions
1. Inclusion and exclusion criteria should be (more) formally cited as well as the primary endpoint of the study.
2. Functional diarrhea is not associated with high calprotectin level, and this biological characteristic is usually used to exclude functional disorder and consider (or even diagnose) inflammatory bowel disease (IBD). This is clearly stated in the reference proposed by authors to sustain their research strategy. Please explain.
3. When and how was calprotectin level tested before inclusion? if yes, please provide data.
4. The text mentions « underlying diseases » while usually this situation is usually an exclusion criterion for studies with FD. Please correct or comment.
5. About 70% of the subjects were males. Please discuss whether this reflects FD in your country (to my knowledge, this is not the case worldwide). Could this influence the study results?
6. The symbiotic should be described. Authors provide some information on its microbial contents at the species level. They should provide details at the strain level, and the quantity of FOS
7. The placebo should be described
8. The randomization method should be described.
9. Instead of “synbiotics” the results should be ascribed in the text, title and tables to the specific product tested
10. Introduction: the term “irritable bowel disease” is not adequate. It suggests some confusion between irritable bowel syndrome (and the usual acronym of this situation “IBS” is used by authors just after the term) and inflammatory bowel disease (“IBD”) which are completely different diseases with inflammation, often high calprotectin levels and very different treatment approaches.
11. The comparisons should discuss primarily differences between the symbiotic and placebo groups (and not the “improvement” in each of the groups)
12. Medical scientific societies consider that the evidence of clinical efficacy of synbiotics (and even probiotics and prebiotics) is not so high and guidelines are not often quoting them as a validated medical approach. This contrasts with the references provided by authors to support their research strategy on these compounds. Authors should simply quote this. For example, the evidence from the meta-analysis suggesting that multistrain would be more effective than single strains is debated.
Round 2
Reviewer 2 Report
1. The commercial name of the product should be removed from the title.
2. the authors did not answer the question on what basis the stool calprotectin value of 11.5 mg/kg was considered as elevated. The norm is 50-200 mcg/gram. At most one can agree with the term high or better detectable calprotectin level.
3. I did not find Table S1.
4. The authors did not answer the question how the content of the different strains was determined in the final product. Was it on the basis of the batch or the analysis of the individual components?
5. Limitations of the LEfSe method should be included in the study limitations.
Reviewer 3 Report
Authors properly provide to the reader which strains of probiotics were used in the study. In the discussion, the refer to species or genus (lactobacillus or bifidobacterium) but never to specific strains which may suggest to the reader that they believe that any strain of L. plantarum (for example) has beneficial effect on functional disorders (which is not true)
Line 271 authors write "the strains used in our study are known to be effective in improving diarrhea and IBS-D" and should provide here a reference
Authors compared variables of intestinal functioning such as stool consistency before and after the end of each of the two treatments. I do not share the author interpretation that the symbiotic improved the degree of formed stools as this degree was strictly similar to that of the placebo group at the end of the study.
